# Mesenchymal Stem Cells and Formyl Peptide Receptor 2 Activity in Hyperoxia-Induced Lung Injury in Newborn Mice

**DOI:** 10.3390/ijms231810604

**Published:** 2022-09-13

**Authors:** Young Eun Kim, So Yoon Ahn, Dong Kyung Sung, Yun Sil Chang, Won Soon Park

**Affiliations:** 1Cell and Gene Therapy Institute, Samsung Medical Center, Seoul 06351, Korea; 2Department of Health Sciences and Technology, SAIHST, Samsung Medical Center, Seoul 06351, Korea; 3Department of Pediatrics, Samsung Medical Center, Sungkyunkwan University School of Medicine, Seoul 06351, Korea

**Keywords:** mesenchymal stem cell transplantation, bronchopulmonary dysplasia, formyl peptide receptor 2, newborn, diseases

## Abstract

Formyl peptide receptor (FPR) 2 is known to play a critical role in regulating inflammation, including either the pro-inflammatory or pro-resolving effects. However, its role in neonatal hyperoxia-induced lung injury has not been delineated. In this study, we investigate whether mesenchymal stem cells (MSCs) attenuate hyperoxia-induced neonatal lung injury by regulating FPR2 activity. We observed a significant increase in FPR2 levels in alveolar macrophages (RAW264.7 cells) after H_2_O_2_-induced stress, which decreased after MSC treatment. In the H_2_O_2_-induction model, increased levels of inflammatory cytokines (IL-1α and TNF-α) were significantly reduced in RAW264.7 cells after treatment with WRW4, an inhibitor of FPR2, or MSCs. Viability of lung epithelial cells and endothelial cells was significantly improved when cultured in the conditioned media of RAW264.7 cells treated with WRW4 or MSCs, compared to when cultured in the conditioned media of control RAW265.7 cells exposed to H_2_O_2_. For the in vivo study, wild-type and FPR2 knockout (FPR2^−/−^) C57/BL6 mouse pups were randomly exposed to 80% oxygen or room air from postnatal day (P) 1 to P14. At P5, 2 × 10^5^ MSCs were transplanted intratracheally. MSCs reduced the elevated FPR2 activity at P7 and improved the decreased FPR2 activity as well as the increased immuno-stained FPR2 activity in alveolar macrophages in hyperoxic lungs at P14. Both FPR2^−/−^ and MSCs similarly attenuated impaired alveolarization and angiogenesis, and increased apoptosis and inflammation of hyperoxic lungs without synergistic effects. Our findings suggest that the protective effects of MSCs in hyperoxic lung injury might be related to indirect modulation of FPR2 activity, at least of alveolar macrophages in neonatal mice.

## 1. Introduction

Bronchopulmonary dysplasia (BPD) is a common complication in preterm infants that requires prolonged mechanical ventilation and oxygen supplementation. Hyperoxia-induced lung injury is a major cause of mortality and morbidity among preterm infants. However, despite many advances in neonatal intensive care, effective treatments have not yet been developed. In recent years, the anti-inflammatory effects of mesenchymal stem cells (MSCs) and their mechanisms of action in various inflammatory diseases have attracted attention. Previously, we have proved that the intratracheal transplantation of human umbilical-cord, blood-derived MSCs significantly attenuated hyperoxia-induced lung injuries in a newborn rodent model [1,2]. The therapeutic mechanism of MSCs is attributed to their anti-inflammatory, anti-apoptotic, and angiogenic effects via paracrine signaling [3]. In the phase I clinical trial, the clinical safety and feasibility of the transplanted MSCs in preterm infants at risk for developing BPD were investigated and followed up to two years of corrected age [4,5].

FPR2 is a highly versatile receptor due to its ability to bind various ligands, such as peptides, proteins, and lipids [6]. FPR2 can trigger both pro-inflammatory and anti-inflammatory pathways depending on ligands. According to previous studies, serum amyloid A could promote neutrophilic inflammation via FPR2 in chronic obstructive pulmonary disease [7]. Imbalance between pro-inflammatory and anti-inflammatory ligands of FPR2 can promote inflammation in inflammatory diseases [8,9]. However, the role and expression pattern of FPR2 in controlling inflammation has not been elucidated in neonatal hyperoxic lung injury, and a better understanding of the molecular mechanism of the MSC action is important for their future application in clinical care.

The formyl peptide receptor (FPR) 2, which is mainly expressed in phagocytes, plays an important role in host defense and inflammation by regulating cell recruitment and activation [10]. FPR2 interacts with pro-inflammatory ligands, such as mitochondrial and bacterial formylated peptides, in various diseases [11]. It is also known to interact with pro-resolving ligands such as lipoxin A4, resolvin D1, and the synthetic peptide WKYMVm, which have been shown to ameliorate hyperoxic lung injuries in animal models [12,13,14]. Therefore, targeting FPR2 may alter and prevent the pathogenesis of inflammatory lung diseases. However, FPR2′s role in MSCs’ therapeutic mechanism in hyperoxia-induced BPD has not been studied yet.

Therefore, in this study, we primarily investigate the role of FPR2 signaling in the development of hyperoxia-induced lung injury resulting in lung inflammation, cell death, and impaired alveolarization and angiogenesis. Second, we investigate the regulatory effect of the transplantation of MSCs in the context of FPR2 signaling using an in vitro model of H_2_O_2_-induced stress-exposed alveolar macrophages, lung epithelial and endothelial cell lines, and an in vivo model of hyperoxia-exposed wild-type and FPR2^−/−^ neonatal mice.

## 2. Results

### 2.1. The Effect of MSCs and WRW4 on RAW264.7 Cells

Since WRW4, an FPR2 antagonist, did not show a significant effect on the viability of L2 lung epithelial cells in an H_2_O_2_-treated single-culture condition (Appendix A), we evaluated WRW4’s effect on regulating activation of RAW264.7 alveolar macrophages for the next step (Figure 1). After H_2_O_2_-induction, the level of FPR2 significantly increased in RAW264.7 cells compared to non-treated normal control (Figure 1A). However, WRW4 and MSC treatments significantly reduced the increase in FPR2 level on H_2_O_2_-induced RAW264.7 cells. The levels of inflammatory cytokines, such as IL-1α and TNF-α, also increased significantly in H_2_O_2_-induced RAW264.7 cells, but significantly reduced after WRW4 and MSC treatments (Figure 1B). The WRW4 and MSCs did not have a synergistic effect on the reduction of inflammatory response of RAW264.7 cells. To indirectly investigate level of an endogenous source of FPR2 activator, we measured the level of extracellular mitochondrial DAMP (mitochondrial DNA) released in conditioned media of H_2_O_2_-exposed RAW264.7 macrophages (Appendix A). The level of mitochondrial DAMP significantly increased after H_2_O_2_ exposure compared to the normal control group but significantly reduced when co-cultured with MSCs and treated with WRW4, compared to the H_2_O_2_ control group. However, no synergistic effect was observed between MSCs and WRW4 in this experiment.

Thereafter, we observe the viability of L2 and HULEC-5a cells incubated in conditioned media of RAW264.7 cells (Figure 1C); prior to transferring the media from RAW264.7 cells to L2 or HULEC-5a cells, RAW264.7 cells were activated by H_2_O_2_-induction and treated with WRW4 or MSCs or were non-treated. The viability of L2 significantly reduced after incubation in conditioned media of H_2_O_2_-induced RAW264.7 cells compared to control media of RAW264.7 cells (Figure 1C, left panel). However, the reduced viability of L2 cells significantly improved when the RAW264.7 cells’ FPR2 was inhibited by WRW4 and MSCs. In line with this, a similar result was observed in the viability of HULEC-5a cells (Figure 1C, right panel). In short, the inhibition of FPR2 levels in RAW264.7 cells significantly ameliorated toxicities, such as inflammatory cytokines, which affect L2 and HULEC-5a cell viabilities.

### 2.2. Effects of FPR2^−/−^ and MSCs on Body Growth, and the Level of FPR2 in a Hyperoxic Newborn Mouse

The postnatal day (P) 1 body weights of mice in all the groups were evenly distributed (Figure 2A). Under normoxic conditions, wild-type and FPR2^−/−^ mice showed no significant differences in P1 and P14 body weights (Figure 2A). After hyperoxia exposure, P14 body weight of wild-type mice significantly reduced compared to the wild-type normoxic control. However, the body weight of FPR2^−/−^ mice did not significantly reduce after hyperoxia exposure compared to the FPR2^−/−^ normoxic control, and the body weight of hyperoxia-exposed FPR2^−/−^ mice was significantly higher than that of hyperoxia-exposed wild-type mice. The intratracheal transplantation of MSCs significantly improved the hyperoxia-induced reduction in the body weight of wild-type mice. The body weight of FPR2^−/−^ mice did not change significantly after MSC transplantation.

The level of FPR2 was measured at P7 and P14 in wild-type mice, and the trend of change in FPR2 levels was different in the hyperoxia control group compared to the normal control group (Figure 2B). The level of FPR2 in the hyperoxic control group at P7 significantly increased, but it significantly reduced at P14 compared to the normal control group. However, a reverse trend was observed in the changes in FPR2 levels at P7 and P14 after MSC transplantation.

### 2.3. Effects of FPR2 Knockout and MSCs on Alveolarization of Hyperoxic Lungs

Under normoxic control conditions, there was no significant differences in lung alveolarization between the wild-type and FPR2^−/−^ mice at P14. Hyperoxia caused alveolarization impairment with larger, fewer, and more heterogenous sizes of alveoli than the normoxia control at P14 (Figure 3A). The mean linear intercept (MLI) and mean alveolar volume (MAV) were evaluated to quantify the degree of alveolarization (Figure 3B). After exposure to hyperoxia, MLI and MAV significantly increased in both wild-type and FPR2^−/−^ mice compared to the normoxic controls, but the MLI and MAV in FPR2^−/−^ mice were significantly lower than those in wild-type mice. In wild-type mice, MSC transplantation significantly improved hyperoxia-induced alveolarization damage compared to the hyperoxic control. However, in FPR2^−/−^ mice, there were no significant differences in the MLI and MAV between the hyperoxic control and MSC–transplanted groups.

### 2.4. Effects of FPR2 Knockout and MSCs on Angiogenesis and Cell Death of Hyperoxic Lungs

Impaired angiogenesis is another feature of neonatal hyperoxia-induced lung injury. There were no significant differences in lung angiogenesis between normoxic controls of wild-type and FPR2^−/−^ mice at P14. Exposure to hyperoxia impaired lung angiogenesis was seen as a reduced light intensity of von Willebrand factor (vWF) in both wild-type and FPR2^−/−^ mice compared to the normoxic controls at P14 (Figure 4A). When the hyperoxia-exposed lungs of wild-type and FPR2^−/−^ mice were compared, the light intensity of vWF was significantly higher in FPR2^−/−^ mice than in wild-type mice. Hyperoxia-induced impairment in lung angiogenesis in wild-type mice significantly improved after MSC transplantation compared to the hyperoxic control. However, there was no significant differences in vWF light intensity between the hyperoxic control and MSC–transplanted groups in FPR2^−/−^ mice.

The terminal deoxynucleotidyl transferase dUTP nick end labeling (TUNEL) assay was used to compare the number of dead lung cells, and no significant differences were found between normoxic controls of wild-type and FPR2^−/−^ mice at P14. Hyperoxia significantly increased the number of TUNEL–positive cells compared to the normoxic control in wild-type and FPR2^−/−^ mice (Figure 4B). When the hyperoxia-exposed lungs of wild-type and FPR2^−/−^ mice were compared, the number of TUNEL–positive cells was significantly lower in FPR2^−/−^ mice than in wild-type mice. The hyperoxia-induced increase in the number of TUNEL–positive cells in wild-type mice significantly attenuated after MSC transplantation compared to the hyperoxic control. However, in FPR2^−/−^ mice, there was no significant difference in the number of TUNEL–positive cells between the hyperoxic control and MSC transplanted group.

### 2.5. Effects of FPR2 Knockout and MSCs on Inflammation in Hyperoxic Lungs

The number of migrated inflammatory cells, including macrophages and leukocytes, marked by CD68 and myeloperoxidase (MPO), respectively, was not significantly different between the normoxic control lungs of wild-type and FPR2^−/−^ mice at P14. Almost no infiltration of CD68– and MPO–positive cells was observed in the normoxic lungs, but exposure to hyperoxia significantly increased the number of CD68– and MPO–positive cells compared to the normoxic controls in wild-type and FPR2^−/−^ mice (Figure 5A,B). When hyperoxia-exposed lungs of wild-type and FPR2^−/−^ mice were compared, the number of CD68– and MPO–positive cells in FPR2^−/−^ mice was significantly lower than that in the wild-type mice. The hyperoxia-induced increase in cell infiltration of wild-type mice significantly attenuated after MSC transplantation compared to the hyperoxic control. However, in FPR2^−/−^ mice, there was no significant difference in the number of CD68– and MPO–positive cells between the hyperoxic control and MSC–transplanted groups. Most of the CD68–positive cells were co-merged with FPR2 (Figure 5C). We estimated the level of FPR2 expressed on the double-stained cells with FPR2 and CD68 at the single-cell level, in wild-type mice at P14 (Figure 5C). The average light intensities of FPR2 (red) co-labelled with CD 68 (green) were enhanced in hyperoxic control group compared to normal control group, and was decreased in the MSC transplanted group compared to hyperoxic control group.

The levels of lung inflammatory cytokines, such as IL-1α, IL-1β, and IL-6, measured in lung tissues at P14 significantly increased after hyperoxic lung injury compared to the normoxic control in wild-type (Figure 5D). On the other hand, in FPR2^−/−^ mice, the levels of inflammatory cytokines did not significantly increase after hyperoxic lung injury compared to the normoxic control. Compared to wild-type mice, FPR2^−/−^ mice showed a significantly lower inflammatory responses induced by hyperoxic lung injury. While MSC transplantation significantly attenuated the levels of IL-1α, IL-1β, and IL-6 in wild-type lungs, there was no additional effect of MSC transplantation in FPR2^−/−^ mice compared to that in FPR2^−/−^ control mice.

## 3. Discussion

We recently proved that intratracheal transplantation of human umbilical cord blood-derived MSCs has therapeutic effects mainly by immune modulating effects in neonatal rodent models of hyperoxia-induced lung injury [1,2]. Then, in lung injury, FPR2 was well known to be related to the macrophage induced inflammation [10]. We thus attempted to investigate whether the protective effects of MSCs is altered with FPR2 inhibition in the hyperoxic lung injury using FPR2^−/−^ and wild-type neonatal mice.

First, we demonstrated that FPR2 inhibition significantly attenuated hyperoxia-induced lung injury both in vitro and in vivo. The viability of H_2_O_2_-exposed lung epithelia (L2 cells) was not significantly changed after WRW4 treatment compared to the non-treated group. However, when L2 cells were incubated with conditioned media of alveolar macrophages (RAW264.7 cells), the L2 cell viability was changed depending on FPR2 inhibition in RAW264.7 cells; when FPR2 of RAW264.7 cells was inhibited, L2 cell viability was higher than when FPR2 of the cells was not inhibited. After FPR2 inhibition, inflammatory cytokine levels were reduced in RAW264.7 cells. This suggests that inhibition of FPR2 might protect lung cells by suppressing macrophage activation and related cytokine secretion in vitro and in vivo models. In the in vivo study using neonatal hyperoxic lung injury model, we confirmed that the lung of FPR2^−/−^ showed enhancements in alveolarization and angiogenesis as well as reductions in apoptosis, infiltrated inflammatory cells, and levels of inflammatory cytokines, compared to wild-type mouse lungs.

Second, we demonstrated that MSCs attenuated neonatal hyperoxic lung injury by indirectly modulating FPR2 expression level, at least in alveolar macrophages. In RAW264.7 alveolar macrophages exposed to H_2_O_2_-stress, mitochondrial DNA release, FPR2 expression level, and pro-inflammatory cytokine production were increased, but they reduced after MSC treatment. The in vivo study recreated our previous experiment to observe the therapeutic effect of intratracheally transplanted MSCs on improving alveolarization and angiogenesis, as well as on reducing inflammatory cell infiltration, inflammatory cytokine levels, and apoptosis, in a hyperoxic neonatal rodent model of wild-type [1,2]. In this model, FPR2 activities were higher initially (at 7 days) followed by lower activity later (at 14 days of age), and MSC transplantation reversed these trends showing lower FPR2 activity initially, followed by higher FPR2 activity than that of the hyperoxic control (Figure 2B) accompanied by attenuated hyperoxic lung injuries. In our hyperoxic lung injury model, the histological phenotypes of the MSC–transplanted group of wild-type mice were almost the same as those of the non-transplanted control group of FPR2 knockout mice. Meanwhile, there was no significant difference between the control group and the MSC–transplanted group in hyperoxia-exposed FPR2 knockout mice. This suggests that FPR2 regulation is important in hyperoxic lung injury but MSCs may, indirectly rather than directly, attenuate lung injury via FPR2 signaling.

According to previous studies, FPR2 played an essential role in progression of LPS–induced acute lung injury by increasing levels of oxidative stress and pro-inflammatory cytokines in macrophages [15], and FPR2 knock-downed macrophages showed a lower inflammatory response compared to controls [15,16]. However, the involvement of FPR2 on macrophages in neonatal hyperoxia-induced lung inflammation has not been studied. In our present study, we indirectly investigated that release of mitochondrial DAMP, a source of FPR2 ligand, reduced after co-cultured with MSCs and treated with FPR2 inhibitor, WRW4, in alveolar macrophages. Then, we observed that increased levels of FPR2 and inflammatory cytokine (IL-1α and TNFα) in H_2_O_2_ induced-alveolar macrophages were significantly reduced after co-cultured with MSCs or treated with FPR2 inhibitor (WRW4) to a similar extent, in alveolar macrophages. In vivo study confirmed that MSC transplantation and FPR2 deficiency reduced hyperoxia-induced lung inflammation to a similar extent. Furthermore, in our hyperoxic lung injury of in vivo model, increased FPR2 activity evidenced by increased light intensity of CD68 stained alveolar macrophage was attenuated by MSC transplantation. It might suggest that alveolar macrophages participate in hyperoxia-induced lung inflammation via FPR2, and MSCs attenuated the hyperoxia-induced lung injury by indirectly modulating FPR2 levels in alveolar macrophages.

FPR2 is a versatile transmembrane protein belonging to the G-protein coupled receptor family and is mainly expressed on monocytes and neutrophils [17]. Lipoxin A4, annexin A1, resolvin D1, and synthetic peptide WKYMVm, which are pro-resolving mediators, and attenuate inflammatory reactions via FPR2 [18]. Previous reports have shown that lipoxin A4 and resolvin D1 do not function properly in FPR2 knockout mice [19,20] and that WKYMVm regulates inflammatory reactions and reduces hyperoxic lung injury [14,21]. In contrast, FPR2 signaling can be activated in the opposite direction to the pro-inflammatory response depending on different types of stimulation, such as by LL-37 and mitochondrial formyl peptides [22]. According to another report, FPR2 deficiency relieved inflammation by suppressing M1 polarization and inflammation mediated by macrophage chemotaxis in obesity-related inflammation and atherosclerosis [16,23]. In this study, we observed that the level of FPR2 in the hyperoxic lung was increased at P7 but reduced at P14 compared to the normal control (Figure 2B), whereas the level of FPR1 constantly increased at P7 and P14 compared to the normal control (Appendix A). Unlike FPR1, FPR2′s role and its expression level might depend on the phase of initiation and resolution of inflammation and target cells [16,24]. However, FPR2 has similar properties to FPR1 in that it binds to formylated peptides derived from the bacteria and mitochondria of injured cells, recruits neutrophils and macrophages, and has a negative effect on promoting inflammation. We observed that levels of FPR1 and FPR2 significantly increased in RAW264.7 alveolar macrophages in an in vitro model of H_2_O_2_-stress but reduced after MSC treatment (Figure 1A and Appendix A), and that the levels of FPR1 and FPR2 in hyperoxic lungs were normalized after MSC transplantation (Figure 2B and Appendix A). In the in vivo study using knockout mice of FPR1 and FPR2, the pathological phenotypes of hyperoxia-induced lung injury, such as impairment of alveolarization, inflammation, and cell death, significantly attenuated compared to wild-type mice [2]. Considering the functional knockout of FPR2 and FPR1 showed a similar tendency to attenuate hyperoxia-induced lung injury, we speculate that the knockout of FPR2, as well as FPR1, would contribute to protecting lung injury from inflammatory progression. This might suggest that control of gene expression of FPR2 is important for immune modulation, and if the gene expression of FPR2 cannot be controlled, FPR2 could increase inflammation in hyperoxic lung injury in newborn rats.

In hyperoxic lung injury, inflammatory cells, such as macrophages, might be activated via FPR2, and the subsequent production of cytokines and free radicals can promote inflammation, damage healthy cells, and impair lung development in newborns [25]. It may be suggested that normalized levels of FPR2 after MSC transplantation improve lung inflammation. According to a related report, significantly higher levels of FPR2 were observed in umbilical cord blood and placenta obtained from pregnant women with preeclampsia [26]. Increased levels of FPR2 in sputum cells were also observed in intermittent asthma, whereas a reduced level of FPR2 was observed in severe asthma compared to healthy controls. This suggests that preventing lung inflammation via knockout of FPR2 would help protect the lung from uncontrolled inflammation and alleviate tissue damage from hyperoxia-induced lung injury, but controlling FPR2 levels in hyperoxia-induced lung injury of newborns should be carefully considered due to FPR2′s versatile functions depending on the stage of the disease and ligand-specific signaling. However, there are several limitations in the present study. We did not measure N-formyl peptide directly in the present study. Instead, we measured the level of extracellular mitochondrial DNA by qPCR to investigate extracellular mitochondrial DAMP in conditioned media of H_2_O_2_-exposed RAW264.7 macrophages (Appendix A). The level of mitochondrial DNA release measured in extracellular media significantly increased after H_2_O_2_ exposure compared to normal control group but significantly reduced when co-cultured with MSCs and treated with WRW4, which is an antagonist of FPR2, compared to the H_2_O_2_ control group. According to Ben Lu [27], H_2_O_2_ stress induces mitochondrial DNA release into the cytoplasm in mouse macrophages and activates inflammasome. The only source of endogenous N-formyl peptide is a mitochondrial release from injured or dead cells [28]. We did not measure the level of the direct ligand of FPR2, such as N-formyl peptide, but our study suggests that MSCs and FPR2 inhibition downregulate mitochondrial DAMP–related inflammatory response in macrophages induced by oxidative stress. Although we did not directly investigate the direct link between FPR2 and its ligand, such as N-formyl peptide in vitro and in vivo, our study might suggest that MSCs and FPR2 inhibition downregulate mitochondrial DAMP–related inflammatory response in macrophages after H_2_O_2_ induced-stress. Additionally, we used H_2_O_2_-induced stress in an in vitro model in this study, but prolonged 24 h exposure to 95% O_2_ could be a more ideal modeling method to induce hyperoxia-compromised macrophage function. However, previous studies have showed H_2_O_2_ treatment successfully generated oxidative stress-induced cell death and inflammation in pulmonary endothelial cells and epithelial cells which resembles findings induced by prolonged oxygen treatment in vitro [3,29]. Moreover, the previous report proved that H_2_O_2_ developed concentration-dependent oxidative damage evidenced by decreased cell survival rate, increased LDH, and upregulated TNF—a release in the alveolar macrophages [30]. Similarly, our data in the present study also resembles the finding above. The result of our present study displayed increased inflammatory cytokines such as IL-1α, and TNF-α- in the alveolar macrophages (RAW264.7cells) after H_2_O_2_ treatment, which provides supporting evidence that H_2_O_2_ treatment may induce oxidative stress not only in the pulmonary endothelial and epithelial cells, but also in the alveolar macrophages. In our previous report about VEGF knock-down mesenchymal stem cell treatment in the hyperoxic lung injury in an in vitro model [29], the experimental group using VEGF knockdown (VEGF siRNA-transfected) mesenchymal stem cells and the other experimental group using MSCs with a VEGF blocking antibody showed quite similar results in cell death and inflammatory reactions. Given these findings above, in this current study, to mimic the FPR2 knock-downed alveolar macrophages, we pretreated WRW4 to the alveolar macrophages 2 h before MSC co-treatment. However, rather than using a blocking antibody pretreatment, genetically engineered FPR2 knock-down macrophages would be more favorable, but needs future further study. Additionally, we did not delineate how different the role of FPR2 is in acute and chronic phases and initiation and resolution phases in hyperoxia-induced lung injury in newborns. Although the exact change in the signaling mechanism by up- or down-regulation of FPR2 in inflammatory diseases has not yet been identified, FPR2 is considered an important membrane receptor in the regulation of inflammatory cell migration and related injuries. Another limitation of this study is that, although we revealed that MSCs attenuated the inflammatory response by regulating FPR2 levels in alveolar macrophages, we did not directly determine how MSCs regulate the level and which downstream of FPR2 participates in lung injury. We did not measure the levels of pro-inflammatory ligands of FPR2, such as the formyl peptide and serum amyloid A, in our hyperoxic lung injury rodent model. Furthermore, we could not observe any synergistic effect on decreasing injuries by combining MSC transplantation and FPR2 knockout in in vivo experiments. This finding is in line with in vitro results showing that there was no synergistic effect of FPR2 inhibition and MSC treatment on reducing the inflammatory response of alveolar macrophages. However, further studies are needed to investigate more precise mechanisms for this. In addition, in this experiment, changes in pulmonary hypertension, FPR2 level in the heart tissue, and effects of FPR2 over-expressing was not delineated, and the sex variation for the in vivo models was also not investigated, and which need further studies.

In conclusion, we investigated whether the inhibition of FPR2 and administration of MSCs attenuated hyperoxia-induced lung injury in vitro and in vivo. We found that lung epithelial and endothelial cells were injured by FPR2-mediated activation of alveolar macrophages, and that FPR2 knockout mice were significantly protected from hyperoxia-induced lung injury compared to wild-type mice. Furthermore, the protective effects of MSCs transplantation on hyperoxic lung injury were related to the indirect modulation of FPR2 activity, at least of alveolar macrophage in neonatal mice.

## 4. Materials and Methods

### 4.1. Preparation of Mesenchymal Stem Cells

MSCs were isolated from human umbilical-cord blood from a single donor with maternal informed consent from Medipost Co., Ltd. (Seoul, Korea) and cultivated as described previously [1]. Briefly, the cultured cells were maintained at 37 °C in a humidified atmosphere (21% O_2_) containing 5% CO_2_ with a change in culture medium twice a week. MSCs at passage 5–6 were used in this study.

### 4.2. Effect of WRW4 and MSCs on Alveolar Macrophages

Alveolar macrophages (RAW264.7 cells; Korean Cell Line Bank, Seoul, Korea) were cultured in 24-well plates at a density of 5 × 10^4^ cells. For FPR2 inhibition, RAW264.7 cells were treated with WRW4 (10 μM; Tocris, Ellisville, MO, USA), a specific FPR2 inhibitor, for 2 h before H_2_O_2_-induction. In the MSC treatment group, MSCs were co-cultured at a ratio of 5:1 with H_2_O_2_-induction. For the H_2_O_2_-induced oxidative stress assay, cells were exposed to 100 μM H_2_O_2_ for 12 h. Levels of FPR2 were measured in the cell lysates by western blotting using FPR2 antibodies (NLS1878, Novus Biologicals, Littleton, CO, USA), and levels of inflammatory cytokines, such as IL-1α and TNF-α, were measured in cell-conditioned media by enzyme-linked immunosorbent assay (R&D Systems, Minneapolis, MN, USA) according to the manufacturer’s protocol.

### 4.3. In Vitro Cell Survival Assay

The effects of MSCs and WRW4, a FPR2 antagonist, on the viability of lung epithelial cell lines (L2 cells, Korean Cell Line Bank, Seoul National University College of Medicine, Seoul, Korea) and lung endothelial cell lines (HULEC-5a cells, American Type Culture Collection, Manassas, VA, USA) were measured using a Cell Counting Kit-8 assay (CCK-8; Dojindo, Kumamoto, Japan). L2 and HULEC-5a cells were cultured in 96-well plates at a density of 5 × 10^4^ cells/well. The conditioned media of alveolar macrophages (RAW264.7 cells, Korean Cell Line Bank) with or without co-culture along with MSCs or treatment of WRW4 was centrifuged at 300 g at 4 °C for 10 min to settle the cells, which were then transferred to culture plates of L2 or HULEC-5a cells. Viabilities of L2 and HULEC-5a cells were observed after incubation with conditioned media of H_2_O_2_-induced RAW264.7 alveolar macrophages. The RAW264.7 cells were untreated, or treated with WRW4 (10 μM; Tocris, Ellisville, MO, USA) and MSCs (co-cultured in ratios of 5:1) under the hypothesis that the conditioned medium composition of RAW264.7 cells, which affects the viability of L2 and HULEC-5a cells, would be different after FPR2 inhibition and MSC treatment.

### 4.4. Measurement of Mitochondrial DNA Level

Extracellular mitochondrial DNA in RAW264.7 cell culture media was extracted using DNeasy Blood and Tissue kits (QIAGEN, Hilden, Germany), according to the manufacture’s protocols. Level of mitochondrial DNA in the cell-free culture media was measured by PCR using AccuPower PCR PreMix (Bioneer, Seoul, Korea) and PCR amplification of rat mtDNA was performed in the following conditions: 5 min hot start at 94 °C; followed by 33 cycles of 94 °C for 30 s, 60 °C for 30 s, 72 °C for 30 s; and a final extension at 72 °C for 5 min. The primer sequences for rat mtDNA were as follows: forward 5′-AGGACTTAACCAGACCCAAACACG-3′, reverse 5′-CCTCTTTTCTGATAGGCGGG-3′. The PCR products were resolved and visualized by E-Gel Power Snap Electrophoresis System (Invitrogen). PCR band intensity of rat mtDNA was measured using ImageJ Version 1.51 software (National Institutes of Health, Bethesda, MD, USA).

### 4.5. Animal Model of Hyperoxia-Induced Lung Injury

The animal experiments were reviewed and approved by the Institutional Animal Care and Use Committee of Samsung Biomedical Research Institute (SBRI) (Seoul, Korea). SBRI is an Association for Assessment and Accreditation of Laboratory Animal Care International (AAALAC International)-accredited facility and abides by the Institute of Laboratory Animal Resources guide. The experimental procedures were performed in accordance with the National Institutes of Health Guidelines for Laboratory Animal Care. All animal procedures were performed at an AAALAC-accredited animal care facility. Heterotypes of FPR2 mice (FPR2^+/−^) were kindly gifted by Prof. Jae ho Kim, Pusan National University, Yangsan, Korea. We generated wild-type and FPR2 knockout (FPR2^−/−^) mice by mating the transgenic mice and confirmed their genotypes using polymerase chain reaction (PCR) following the instructions of the laboratory that developed the knockout mice (Appendix A) [10]. Timed-pregnant mice were maintained under a 12 h light/12 h dark-light-dark cycle with constant humidity and temperature in the animal care facility. After birth, wild-type and FPR2^−/−^ mice were randomly grouped according to normoxia (21% oxygen) or hyperoxia (80% oxygen) exposure from postnatal day (P) 1 to P14. The hyperoxic mice were then randomly grouped again according to intratracheal transplantation of MSCs (2 × 10^5^ cells/ 20 μL) or the same volume of vehicle (normal saline) at P5, as previously reported [2]. The mouse pups were reared with dam mice in individual cages (eight pups with a dam mouse in a cage). Six to eight animals per group were used for every read-out in the histological and biochemical analyses. No deaths occurred during any of the animal procedures.

### 4.6. Tissue Preparation

At P14, mouse lungs were obtained following deep pentobarbital anesthesia (60 mg/kg, i.p.) and transcardiac perfusion with ice-cold normal saline. The extracted lung tissues were prepared for each of the following experiments as previously described [8]. Briefly, for histological analyses, the extracted lung tissues were inflated with normal saline at a constant pressure of 20 cm H_2_O, and then immersed in 10% buffered formalin. The inflated and fixed lung tissues were embedded in paraffin and sectioned at 4 μm for histological staining and observation. For biochemical analyses, the extracted lung tissues were immediately snap-frozen, stored at −80 °C, and homogenized prior to the biochemical experiments.

### 4.7. Lung Morphometry

To morphometrically measure lung alveolarization, paraffin-sectioned lung tissues were stained with hematoxylin and eosin and microphotographed at a magnification of 200× (EVOS FL Auto microscope, ThermoFisher Scientific, Waltham, MA, USA). The mean linear intercept (MLI; mean interalveolar distance) and mean alveolar volume (MAV) were measured as described in our previous study [31].

### 4.8. Immunohistochemical Analysis

For histological evaluation of lung angiogenesis, paraffin-sectioned lungs were stained with vWF antibody (ready to use; IR527, FLEX, Dako, Glostrup, Denmark). The vWF-stained areas were photographed under a microscope at 100× magnification. The light intensity of the vWF-positive areas was measured using ImageJ software (National Institutes of Health). To histologically evaluate lung inflammation, paraffin-sectioned lungs were stained with a CD68 antibody (1:100; ab31630, Abcam, Cambridge, UK) and MPO antibody (1:25; ab9535, Abcam). The number of CD68– and MPO– positive cells was counted under a microscope at 200× magnification. In these immunohistochemical analyses, six non-overlapping fields in each stained lung tissue were scanned using a confocal laser scanning microscope (LSM 700, Zeiss, Oberkochen, Germany) by blinded observers. To observe FPR2 expression on alveolar macropahges, FPR2 and CD68 were double-stained with FPR2 antibody (NLS1878; Novus Biologicals, Littleton, CO, USA) and CD68 antibody in wild-type lung tissues. We randomly captured the co-merged cells (n = 6, 10, 11 in WT-NC, WT-HC and WT-HM group) under the microscope at 400× magnification, and estimated the light intensity of FPR2 expressed on the cells at the single-cell level using Image J software. Three animals per group were used in the double-staining analyses.

### 4.9. Terminal Deoxynucleotidyl Transferase dUTP Nick end Labelling (TUNEL) Assay

The number of dead cells in lung tissues was counted following TUNEL staining according to the manufacturer’s protocol using the DeadEnd Fluorometric TUNEL System kit (G3250; Promega, Madison, WI, USA). The TUNEL-stained lung area was counter-stained with 4′,6-diamidine-2′-phenylindole dihydrochloride (DAPI). The number of TUNEL-positive cells was counted in six non-overlapping fields after scanning with a confocal laser scanning microscope (LSM 700, Zeiss, Oberkochen, Germany) by blinded observers.

### 4.10. Enzyme-Linked Immunosorbent Assay

Inflammatory cytokine levels were measured using commercial enzyme-linked immunosorbent assay (ELISA) kits (R&D Systems, Minneapolis, MN, USA) according to the manufacturer’s protocol.

### 4.11. Western Blot

The levels of FPR1 and FPR2 were measured by western blotting. Membranes were blocked and incubated with primary antibodies against FPR1 and FPR2 (NB100-56473, NLS1878; Novus Biologicals). A level of glyceraldehyde-3-phosphate dehydrogenase (GAPDH) (sc-25778; Santa Cruz Biotechnology, Santa Cruz, CA, USA) was used as a loading control. Protein signals were detected using ECL Prime western blotting detection reagent (GE Healthcare, Piscataway, NJ, USA) and Amersham Imager 600 (GE Healthcare Life Sciences, Pittsburg PA, USA). The detected band intensities were measured using the ImageJ software (National Institutes of Health, Bethesda, MD, USA) and the probing protein/GAPDH ratio was calculated from the band intensities.

### 4.12. Statistical Analyses

Data are presented as mean ± standard error of the mean (SEM). Statistical comparisons between groups were evaluated using one-way analysis of variance (ANOVA) and Tukey’s post hoc analysis. All data were analyzed using SAS software (version 9.4; SAS Institute, Cary, NC, USA), and *p*-values less than 0.05 were considered statistically significant.

## Figures and Tables

**Figure 1 ijms-23-10604-f001:**
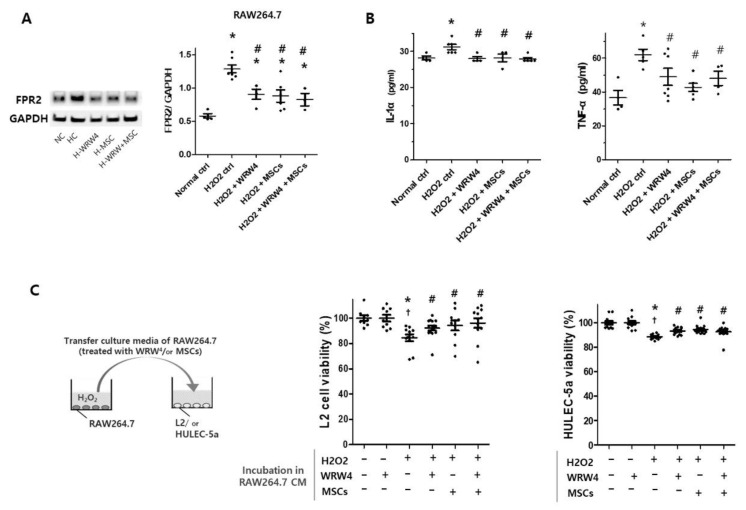
The effects of MSCs and FPR2 inhibitor, WRW4, on RAW264.7 cells’ inflammatory reaction. (**A**) Levels of FPR2 normalized to GAPDH loading control (Full length Western blots are shown in Appendix A) and (**B**) inflammatory cytokines, such as IL-1α and TNF-α, were measured in RAW264.7 cells, 12 h after H_2_O_2_ (100 μM) induction, with or without WRW4 (10 μM) and MSC treatments (RAW264.7 cells: MSC ratio of 5:1). * *p* < 0.05 vs. control group. ^#^ *p* < 0.05 vs. H_2_O_2__-_treated group. (**C**) Viabilities of L2 lung epithelial cells and HULEC-5a lung endothelial cells were measured after incubation in conditioned media (CM) of H_2_O_2_-induced RAW264.7 cells (100 μM of H_2_O_2_ for 12 h), treated with WRW4 (10 μM) and MSCs (RAW264.7 cells: MSC ratio of 5:1), or non-treated. Data are given as mean ± SEM. * *p* < 0.05 vs. normal control group. ^†^ *p* < 0.05 vs. normoxia with WRW4-treated group. ^#^ *p* < 0.05 vs. H_2_O_2_ control group.

**Figure 2 ijms-23-10604-f002:**
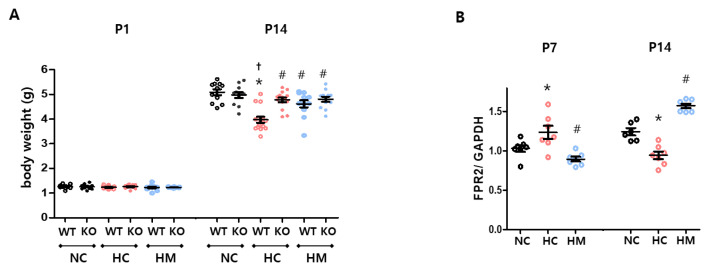
Changes in body weight and levels of FPR2. (**A**) Bodyweight measured at postnatal day 1 (P1; birth weight) and P14 in wild-type and FPR2^−/−^ mice. (**B**) Level of FPR2 normalized to GAPDH loading control in wild-type mice was measured at P7 and P14 (Full-length Western blots are shown in Appendix A). Data are given as mean ± SEM. * *p* < 0.05 vs. NC in WT. ^†^ *p* < 0.05 vs. NC in KO. ^#^ *p* < 0.05 vs. HC in WT. WT, wild-type mouse; KO, FPR2^−/−^ mouse; NC, normoxia control; HC, hyperoxia control; HM, hyperoxia with intratracheal transplantation of human umbilical cord blood-derived MSC transplantation.

**Figure 3 ijms-23-10604-f003:**
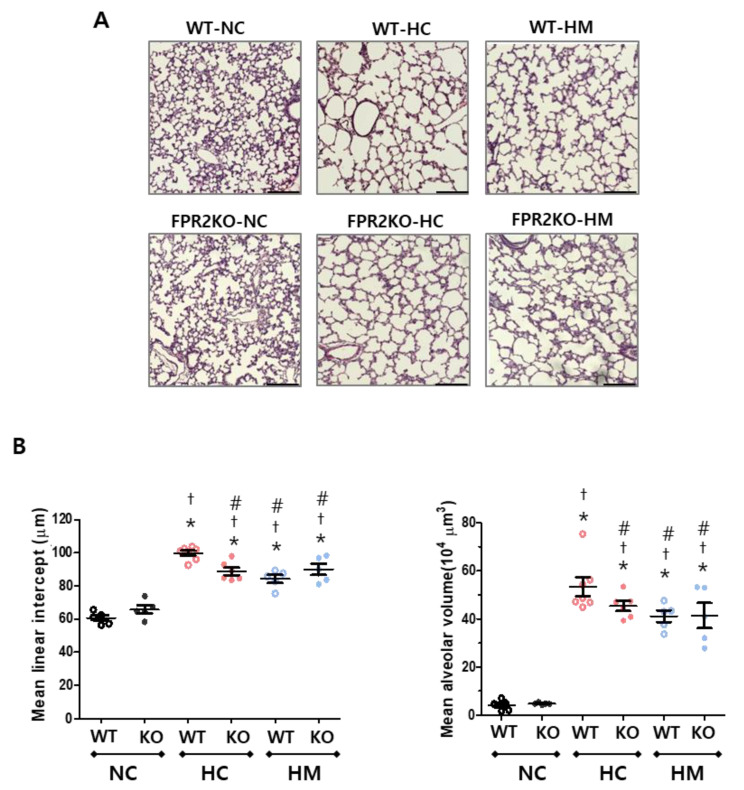
Alveolarization in wild-type and FPR2^−/−^ mice. (**A**) Hematoxylin and Eosin-stained sections showing representative morphology of rat lungs (scale bar, 100 μm). (**B**) Mean linear intercept (μm) and mean alveolar volume (×10^4^μm^3^) as markers of the degree of alveolarization. Data are given as mean ± SEM. * *p* < 0.05 vs. NC in WT. ^†^ *p* < 0.05 vs. NC in KO. ^#^ *p* < 0.05 vs. HC in WT. WT, wild-type mouse; KO, FPR2^−/−^ mouse; NC, normoxia control; HC, hyperoxia control; HM, hyperoxia with intratracheal transplantation of human umbilical cord blood-derived MSC transplantation.

**Figure 4 ijms-23-10604-f004:**
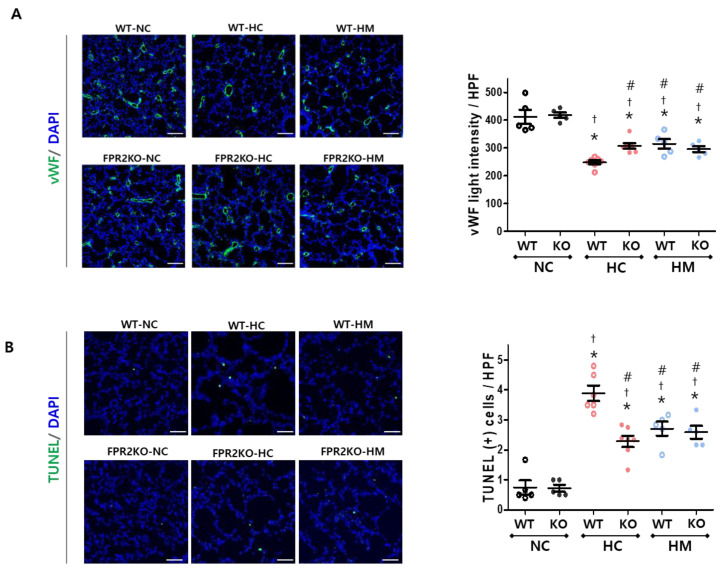
Angiogenesis and cell death in wild-type and FPR2^−/−^ mice. (**A**) Representative photomicrographs of von Willebrand factor (vWF; green) stained lung sections (scale bar, 100 μm) and its light intensity per high-power field (HPF). (**B**) Representative photomicrographs of terminal deoxynucleotidyl transferase dUTP nick end labelling (TUNEL; green) stained lung sections (scale bar, 50 μm) and its quantitative bar graphs. Nuclei were counter-stained with 4′,6-diamidino-2-phenylindole (DAPI; blue). Data are given as mean ± SEM. * *p* < 0.05 vs. NC in WT. ^†^
*p* < 0.05 vs. NC in KO. ^#^ *p* < 0.05 vs. HC in WT. WT, wild-type mouse; KO, FPR2^−/−^ mouse; NC, normoxia control; HC, hyperoxia control; HM, hyperoxia with intratracheal transplantation of human umbilical cord blood-derived MSC transplantation.

**Figure 5 ijms-23-10604-f005:**
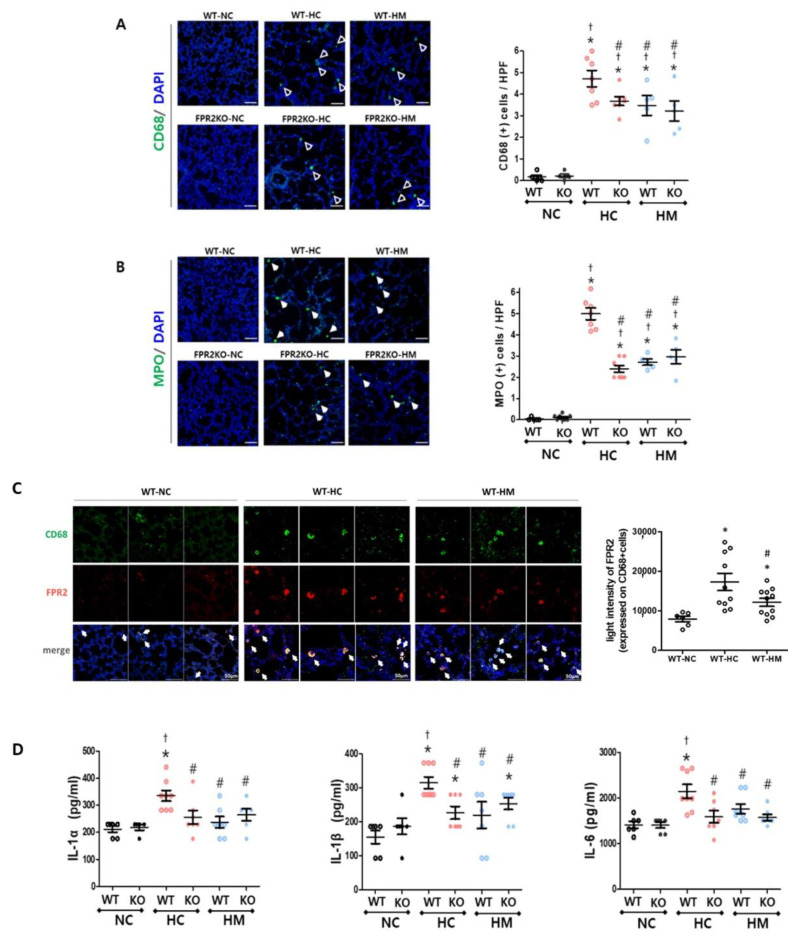
Inflammation in wild-type and FPR2^−/−^ mouse lungs. Representative photomicrographs of (**A**) cluster of differentiation 68 (CD68; green) and (**B**) myeloperoxidase (MPO; green) with respective quantitative bar graphs. Nuclei were counter-stained with 4′,6-diamidino-2-phenylindole (DAPI; blue) (scale bar, 50 μm). (**C**) Representative photomicrographs of double-stained CD68 (green) and FPR2 (red) with quantitative bar graph. (**D**) The levels of interleukin-1α (IL)-1α, IL-1β, and IL-6 measured by enzyme linked immunosorbent assay (ELISA). Data are given as mean ± SEM. * *p* < 0.05 vs. NC in WT. ^†^ *p* < 0.05 vs. NC in KO. ^#^ *p* < 0.05 vs. HC in WT. WT, wild-type mouse; KO, FPR2^−/−^ mouse; NC, normoxia control; HC, hyperoxia control; HM, hyperoxia with intratracheal transplantation of human umbilical cord blood-derived MSC transplantation.

## Data Availability

Not applicable.

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
