# Peer review of "Mesenchymal Stem Cells and Formyl Peptide Receptor 2 Activity in Hyperoxia-Induced Lung Injury in Newborn Mice"

_ijms, 2022, doi:10.3390/ijms231810604_

Round 1

Reviewer 1 Report

This is a very nicely written an presented paper describing the role of the FPR2 in the hyperoxia-induced injury to pulmonary alveolar macrophage cell line in vitro, as well as in vivo.  They demonstrate increase in FPR2 with oxidative stress accompanied by inflammatory responses, and the mitigation of the response by MSCs.  

Comment: With oxidate stress, damage to the mitochondria should occur which could explain the origin of the ligand for the increased formyl peptide receptor, as mitochondria are the only endogenous source of n-formyl peptides.  Would the authors be able to assay the extracellular media or serum for extracellular mitochondria (e.g. by qPCR) or n-formyl peptide levels (e.g. by ELISA) to demonstrate the ligand of the increased FPR2.  There is no correct result, levels may go up, go down, or not change, but this quick experiment would make the paper stronger and would better complete the story of tissue injury, endogenous innate immune activation, and regulation of the response by MSCs.

Author Response

Reviewer #1

  • The reviewer stated, “With oxidate stress, damage to the mitochondria should occur which could explain the origin of the ligand for the increased formyl peptide receptor, as mitochondria are the only endogenous source of n-formyl peptides. Would the authors be able to assay the extracellular media or serum for extracellular mitochondria (e.g. by qPCR) or n-formyl peptide levels (e.g. by ELISA) to demonstrate the ligand of the increased FPR2. There is no correct result, levels may go up, go down, or not change, but this quick experiment would make the paper stronger and would better complete the story of tissue injury, endogenous innate immune activation, and regulation of the response by MSCs.”

-> We appreciate the reviewer’s valuable comments and suggestions. in this experiment, we could not measure N-formyl peptide because there is no N-formyl peptide antibody for ELISA, and HPLC or liquid chromatography needs special equipment and high technical skill for the analyses. Thus, instead of measuring N-formyl peptide level, we measured the level of extracellular mitochondrial DNA by qPCR to investigate extracellular mitochondrial DAMP in conditioned media of H2O2-exposed RAW264.7 macrophages (Supplementary figure S2). The level of mtDNA release measured in extracellular media significantly increased after H2O2 exposure compared to normal control group but significantly reduced when co-cultured with MSCs and treated with WRW4, which is an antagonist of FPR2, compared to the H2O2 control group. According to Ben Lu (New Reference #27; PMID: 24849809), H2O2 stress induces mitochondrial DNA release into the cytoplasm in mouse macrophages and activates inflammasome. We did not measure the level of the direct ligand of FPR2, such as N-formyl peptide, but our study suggests that MSCs and FPR2 inhibition downregulate mitochondrial DAMP-related inflammatory response in macrophages induced by oxidative stress. We incorporated these results into the “Result” section (line 85-93) as Supplementary figure S2 and “Discussion” section in the revised manuscript (line 339-355).

Reviewer 2 Report

Kim et al. 2022 wanted to study the role of mesenchymal stem cells and formyl peptide receptor 2 activity in hyperoxia-induced lung injury in newborn mice reaching a results similar to their previous paper about formyl peptide receptor 1.

This paper is interesting but, in my opinion, lacks to clarity. It would be enough to be more precise in some points to make the reading more fluid.

For examples:

-improve the conclusion of the abstract

-line 52: better explain how FPR2 works in BPD

-Figures: be clearer in the figure legend (e.g., in Fig1A, what is NC or HC...; what is H2O2 or MSCs concentrations...)

-Results: add more details when you explain the pictures for the reason why when I see a figure I already know what I expect in all its aspects without surprise 

-Discussion: try, with the literature in your possession, to postulate what it may be the mechanism by which FPR2 control hyperoxia-induced lung inflammation and why MSC do not choose this way.

Finally, I have some question for my curiosity:

1. why have you chosen the same concentration of your previous study? To have a comparison from the effect on FPR1 and FPR2? Have you try to use other concentrations? we know that MSC could have different action in different concentration.

2.  in your experiments, have you ever though about using MSC's exosomes?

Thanks to your works, I appreciate

Kind Regards

Author Response

  • Reviewer #2

The reviewer stated, “Kim et al. 2022 wanted to study the role of mesenchymal stem cells and formyl peptide receptor 2 activity in hyperoxia-induced lung injury in newborn mice reaching a results similar to their previous paper about formyl peptide receptor 1. This paper is interesting but, in my opinion, lacks to clarity. It would be enough to be more precise in some points to make the reading more fluid.”

For examples:

2-1. improve the conclusion of the abstract

-> As recommended, we have revised a sentence in the conclusion part of the abstract as “Our findings suggest that the protective effects of MSCs in hyperoxic lung injury might be related to indirect modulation of FPR2 activity at least, of alveolar macrophage in neonatal mice.” in the revised manuscript (line 30-32).

2-2. Figures: be clearer in the figure legend (e.g., in Fig1A, what is NC or HC...; what is H2O2 or MSCs concentrations...)

-> As recommended, we have described the experimental group and concentration of H2O2, MSCs and WRW4, in detail in Fig1 and the figure legend (line 106-115).

2-3. Results: add more details when you explain the pictures for the reason why when I a figure I already know what I expect in all its aspects without surprise

-> As recommended, we have added more details in the “Result” section to explain the Figures (line 83, 85-87, 146, 148, 166, 169-170, 179, 200-201 and 218-219).

2-4. Discussion: try, with the literature in your possession, to postulate what it may be mechanism by which FPR2 control hyperoxia-induced lung inflammation and why MSC do not choose this way.

-> According to previous studies, FPR2 played an essential role in progression of LPS-induced acute lung injury by increasing levels of oxidative stress and pro-inflammatory cytokines in macrophages (New Reference #15; PMID: 32106380), and FPR2 knock-downed macrophages showed a lower inflammatory response compared to controls (New Reference #15 and 16; PMID: 32106380 and 25341894). However, the involvement of FPR2 on macrophages in neonatal hyperoxia-induced lung inflammation has not been studied. In our present study, we indirectly investigated that release of mitochondrial DAMP, a source of FPR2 ligand, reduced after co-cultured with MSCs and treated with FPR2 inhibitor, WRW4, in alveolar macrophages. Then, we observed that increased levels of FPR2 and inflammatory cytokine (IL-1α and TNFα) in H2O2 induced-alveolar macrophages were significantly reduced after co-cultured with MSCs or treated with FPR2 inhibitor (WRW4) to a similar extent, in alveolar macrophages. In vivo study confirmed that MSC transplantation and FPR2 deficiency reduced hyperoxia-induced lung inflammation to a similar extent. It might suggest that alveolar macrophages participate in hyperoxia-induced lung inflammation via FPR2, and MSCs attenuated the hyperoxia-induced lung injury by indirectly modulating FPR2 levels in alveolar macrophages. We incorporated this into Discussion section in the revised manuscript (line 276-293)

2-5. line 52: better explain how FPR2 works in BPD

-> FPR2 is a highly versatile receptor due to its ability to bind various ligands, such as peptides, proteins and lipids (New Reference #6; PMID: 33804219). FPR2 can trigger both pro-inflammatory and anti-inflammatory pathway depending on ligands. According to previous studies, serum amyloid A could promote neutrophilic inflammation via FPR2 in chronic obstructive pulmonary disease (New Reference #7; PMID: 23627303). Imbalance between pro-inflammatory and anti-inflammatory ligands of FPR2 can promote inflammation in inflammatory diseases (New Reference #8 and 9; PMID: 24154723 and 25478196). However, the role and expression pattern of FPR2 in controlling inflammation has not been elucidated in neonatal hyperoxic lung injury, and a better understanding of the molecular mechanism of the MSC action is important for their future application in clinical care. We incorporated these into the “Introduction” section (line 51-59).

Reviewer 3 Report

This study seeks to elucidate whether mesenchymal stem cells ( MSCs) attenuate neonatal hyperoxia-induced lung injury by modulating formyl peptide receptor 2 ( FPR2) activity. The authors had previously showed that MSCs down regulate FPR1 activity and this manuscript is an extension of this work. The manuscript is fairly well written however the manuscript is mainly descriptive and there are also key issues that authors need to address.   One of the main questions is that in vitro, they show that H2O2 treated alveolar macrophages have increased FPR2 levels and this was decreased by MSC treatment but there is no mechanistic link. Similarly, in vivo, they show that intratracheal MSCs reduce FPR2 levels in the lungs of hyperoxia-exposed mice, but how the MSCs reduce FPR2 levels is not shown.  

Other major issues:

1. For the in vitro studies, while H2O2 induces oxidative stress, the authors should ideally expose the alveolar macrophages to hyperoxia ( see PUBMED: 24992505). They  should then consider knocking out / overexpressing FRP2 in the alveolar macrophages. 

2) For the data presented in Figure 1C, the authors need to clearly state how the conditioned media was prepared. This is is critical as it is not clear if the endothelial and epithelial cells were a) cultured in the secretome of the  alveolar macrophages (treated with MSCs/WRW) or b) they were cultured in the  secretome of the  alveolar macrophages (treated with MSCs/WRW) and potentially MSC secretome. 

3) Based on the in vitro data presented, the authors suggest that MSC modulation of FRP2 in alveolar macrophages, contribute to MSC beneficial effects on alveolar and vascular structures during hyperoxia. But the link is lost in in vivo. They do show less lung macrophage infiltration with MSC treatment but was FRP2 levels reduced in these macrophages? The authors could isolate macrophages from the bronchoalveolar lavage fluid or do double immunofluorescence staining to put this together.

4) Was there any effect on vascular remodeling or pulmonary hypertension in this model?  Similarly was FRP2 altered in the heart following MSC treatment?

5) The authors show that FRP2 knock out mice have less lung damage and inflammation following hyperoxia exposure and MSCs did not have any synergistic effect. This however does not provide evidence that MSCs attenuate hyperoxia-induced lung injury by down regulating FPR2 activity. What happens if FRP2 is overexpressed in the lung? 

Other minor comments:

1) While the bar graphs are easy to see, the authors need to show dot plots so that the scatter of the data can be seen.

2) The authors should show if there was any sex variation in the response to hyperoxia/MSC for the in vivo studies

Author Response

  • Reviewer #3

3-1.  The reviewer stated, “This study seeks to elucidate whether mesenchymal stem cells (MSCs) attenuate neonatal hyperoxia-induced lung injury by modulating formyl peptide receptor 2 (FPR2) activity. The authors had previously showed that MSCs down regulate FPR1 activity and this manuscript is an extension of this work. The manuscript is fairly well written however the manuscript is mainly descriptive and there are also key issues that authors need to address. One of the main questions is that in vitro, they show that H2O2 treated alveolar macrophages have increased FPR2 levels and this was decreased by MSC treatment but there is no mechanistic link. Similarly, in vivo, they show that intratracheal MSCs reduce FPR2 levels in the lungs of hyperoxia-exposed mice, but how the MSCs reduce FPR2 levels is not shown.

-> We sincerely appreciate the reviewer for giving us insightful comments. In the present study, we could not measure N-formyl peptide level, which is a direct ligand of FPR2, in hyperoxic lungs in the present study due to the limitations of technical skill and lack of special equipment, such as HPLC. Therefore, as we presented above as the answer 1-1, we measured the level of mitochondrial DAMP, such as extracellular mitochondrial DNA (mtDNA), by qPCR in vitro study (Supplementary figure S2). It is known that H2O2 stress induces mitochondrial DNA release into the cytoplasm in mouse macrophages and activates the inflammasome (New Reference #27; PMID: 24849809) and the only source of endogenous N-formyl peptide is a mitochondrial release from injured or dead cells (New Reference #28; PMID: 26067258). As the result, we observed that level of extracellular mtDNA significantly increased in RAW264.7 macrophages after H2O2-induced stress but reduced when co-cultured with MSCs or treated with WRW4, a FPR2 inhibitor. Although we did not directly investigate the direct link between FPR2 and its ligand, such as N-formyl peptide in vitro and in vivo, our study might suggest that MSCs and FPR2 inhibition downregulate mitochondrial DAMP-related inflammatory response in macrophages after H2O2 induced-stress. We have incorporated these into the “Result” section (line 87-93) as Supplementary figure S2 and “Discussion” section in our revised manuscript (line 339-355).

Other major issues:

3-2.  For the in vitro studies, while H2O2 induces oxidative stress, the authors should ideally expose the alveolar macrophages to hyperoxia (see PUBMED: 24992505). They should then consider knocking out / overexpressing FRP2 in the alveolar macrophages.

-> We deeply appreciate reviewer’s insightful comment. We absolutely agree with reviewer’s opinion that, rather than H2O2 stress, prolonged 24h exposure to 95% O2 could be more ideal modeling method to induce hyperoxia-compromised macrophage function. However, unfortunately, due to the inevitable limitation related to restriction of prolonged use of high concentration oxygen in our laboratory facility, we tried to induce oxidative stress with H2O2 treatment. Previous studies have showed H2O2 treatment successfully generated oxidative stress-induced cell death and inflammation in pulmonary endothelial cells and epithelial cells which resembles findings induced by prolonged oxygen treatment in vitro (New Reference #3 and 29; PMID: 24669883 and 29650962). Moreover, the previous report proved that H2O2 developed concentration-dependent oxidative damage evidenced by decreased cell survival rate, increased LDH, and upregulated TNF-a release, in the alveolar macrophages (New Reference #30; PMID: 24164876). Similarly, our data in the present study also resembles the finding above. The result in our present study displayed increased inflammatory cytokines such as IL-1α, and TNF-α- in the alveolar macrophages (RAW264.7cells) after H2O2 treatment. We thus believe H2O2 treatment may induce oxidative stress not only in the pulmonary endothelial and epithelial cells, but also in the alveolar macrophages.

In our previous report about VEGF knock-down mesenchymal stem cell treatment in the hyperoxic lung injury in vitro model (New Reference #29; PMID: 24669883), the experimental group using VEGF knock-down (VEGF siRNA-transfected) mesenchymal stem cells and the other experimental group using MSCs with VEGF blocking antibody showed quite similar results in cell death and inflammatory reactions. Given these findings above, in this current study, to mimic the FPR2 knock-downed alveolar macrophages, we pretreated WRW4 to the alveolar macrophages 2 hours before MSC co-treatment. However, rather than using blocking antibody pretreatment, genetically engineered FPR2 knock-down macrophages would be more favorable which needs future further study and we added this as a limitation of this study in the Discussion section (line 356-378).

3-3.  For the data presented in Figure 1C, the authors need to clearly state how the conditioned media was prepared. This is critical as it is not clear if the endothelial and epithelial cells were a) cultured in the secretome of the alveolar macrophages (treated with MSCs/WRW) or b) they were cultured in the secretome of the alveolar macrophages (treated with MSCs/WRW) and potentially MSC secretome.

-> We appreciate the reviewer’s helpful comments on improving our manuscript. In Figure 1C, the conditioned media of RAW264.7 alveolar macrophages with or without co-culture along with MSCs or treatment of WRW4 was centrifuged at 300 g at 4 °C for 10 min to settle the cells, and then transferred to culture plates of L2 epithelial or HULEC-5a endothelial cells. Therefore, the transferred conditioned media contains the secretome of alveolar macrophages as well as a small amount of WRW4 and the secretome of MSCs. As shown in Supplementary figure S1, we observed that WRW4 did not affect the viability of lung epithelial cells. Moreover, as shown in Figure 1B, we also observed that inflammatory cytokines, such as IL-1α and TNFα, were significantly reduced in conditioned media of H2O2-exposed alveolar macrophages when co-cultured with MSCs, compared to H2O2-exposed control cells, which suggests that the secretome of alveolar macrophages changes after co-culture with MSCs. We incorporated detailed methods of how the conditioned media was prepared into the “Method” section in our revised manuscript as the reviewer suggested. (line 428-431) 

3-4.  Based on the in vitro data presented, the authors suggest that MSC modulation of FRP2 in alveolar macrophages, contribute to MSC beneficial effects on alveolar and vascular structures during hyperoxia. But the link is lost in in vivo. They do show less lung macrophage infiltration with MSC treatment but was FRP2 levels reduced in these macrophages? The authors could isolate macrophages from the bronchoalveolar lavage fluid or do double immunofluorescence staining to put this together.

-> We appreciate the reviewer’s insightful and helpful comments on improving our manuscript. As recommended, we have co-stained FPR2 and CD68, a marker for alveolar macrophages, in wild type mouse lung tissues at P14 in WT-NC, WT-HC, and WT-HM as below. Most of the CD68-positive cells (green) were co-merged with FPR2 (red). We randomly captured the co-merged cells (yellow) and estimated the light intensity of FPR2 (red) expressed on the cells at the single-cell level. As the result, we could observe that the light intensity of FPR2 was significantly enhanced in WT-HC compared to that of the WT-NC group. This increased intensity in WT-HC was significantly diminished in WT-HM. We have incorporated into “Result” section (line 211-217) as Figure 5C in the revised manuscript.

Therefore, it might suggest that a reduced level of FPR2 expressed on alveolar macrophages (Figure 5C in the revised manuscript) as well as reduced numbers of alveolar macrophage infiltration (Figure 5A) in the WT-HM group might contribute to a decrease in inflammation which is associated with attenuating impaired alveolarization and angiogenesis in hyperoxia-induced neonatal lung injuries, suggesting that transplanted MSCs attenuate hyperoxia-induced lung inflammation through modulation of FPR2 signaling not directly, but indirectly via at least in alveolar macrophage.

3-5.  Was there any effect on vascular remodeling or pulmonary hypertension in this model? Similarly, was FRP2 altered in the heart following MSC treatment?

-> We did not observe changes in vascular remodeling, pulmonary hypertension, and FPR2 alteration in the heart in this model. We have incorporated this into the “Discussion” section as one of the limitations of the present study (line 392-395).

3-6.  The authors show that FRP2 knock out mice have less lung damage and inflammation following hyperoxia exposure and MSCs did not have any synergistic effect. This however does not provide evidence that MSCs attenuate hyperoxia-induced lung injury by down regulating FPR2 activity. What happens if FRP2 is overexpressed in the lung?

-> As the reviewer pointed out, we could not observe any synergistic effect on decreasing injuries by combining MSC transplantation and FPR2 knockout in in vivo experiment. This finding is in line with in vitro result showing that there was no synergistic effect of FPR2 inhibition and MSC treatment on reducing the inflammatory response of alveolar macrophages. (Figure 1 A). However, further studies will be needed to clarify the reason for this more precisely, including FPR2 overexpression study in the future. We have incorporated these in the “Discussion” section of our revised manuscript (line 387-395).

Other minor comments:

3-7.  While the bar graphs are easy to see, the authors need to show dot plots so that the scatter of the data can be seen.

-> As recommended, we have presented the data as dot plots instead of bar graphs so that the scatter of the data can be seen in all figures in our revised manuscript.

3-8.  The authors should show if there was any sex variation in the response to hyperoxia/MSC for the in vivo studies.

-> In the present study, we did not observe the sex variation for the in vivo studies. After birth, mouse pups were randomly assigned to NC, HC, and HM group respectively. We have also incorporated this into the “Discussion” section as one of the limitations of the present study (line 394-395)

Round 2

Reviewer 1 Report

Thank you for addressing my experiment.  Using mtDNA qPCR is a great idea and a good substitute for nFP measurement.  I think that these experiments help complete the story and I believe this will be a well-referenced paper.

Reviewer 3 Report

The authors addressed my comments